# *Tropicohilara*, a New Genus of Hilarini (Diptera: Empididae: Empidinae) from Brazil, with Descriptions of Six New Species [note 2]

**DOI:** 10.3390/insects14120915

**Published:** 2023-11-29

**Authors:** Dayse W. A. Marques, Christophe Daugeron, José Albertino Rafael

**Affiliations:** 1Instituto Nacional de Pesquisas da Amazônia (INPA), Coordenação de Biodiversidade, Caixa Postal 2223, Manaus 69060-000, AM, Brazil; willkenia@gmail.com (D.W.A.M.); jarafael@inpa.gov.br (J.A.R.); 2Muséum National d’Histoire Naturelle, Centre National de la Recherche Scientifique, Mécanismes Adaptatifs et Evolution, UMR 7179 MNHN-CNRS MECADEV, CP 50, 45 Rue Buffon, 75005 Paris, France

**Keywords:** systematics, neotropical, Amazon, Atlantic Forest

## Abstract

**Simple Summary:**

A new genus of hilarine dance fly, i.e., *Tropicohilara*, is described with six new included species. The new genus is recorded from the Amazon and Atlantic Forest biomes. The morphology and distribution of the genus are discussed.

**Abstract:**

The genus *Tropicohilara* **gen. nov.** (Diptera: Empididae: Empidinae: Hilarini) is described and includes the following six new species from Brazil: *Tropicohilara amazonensis* **sp. nov.** (type species, Brazil: Amazonas, Manaus); *T. bahiensis* **sp. nov.** (Brazil: Bahia, Camacan); *T. bella* **sp. nov.** (Brazil: Pernambuco, Jaqueira); *T. mineira* **sp. nov.** (Brazil: Minas Gerais, Itamonte); *T. paranaensis* **sp. nov.** (Brazil: Paraná, Piraquara); and *T. sinclairi* **sp. nov.** (Brazil: Paraná, Morretes). The genus is presently recorded from the Amazonian and Atlantic Forest biomes. It differs from other hilarine genera by the following combination of characteristics: predominantly yellowish specimens; occiput somewhat conical in dorsal view; postpedicel elongate, male first fore tarsomere unmodified; hind tibia slightly shorter than hind femur; wing vein R_2+3_ with setae on ventral surface; male tergite 7 with a sclerotized band at posterior margin, tergite 8 reduced and upwardly directed so that terminalia can be flexed forward. A key to the species is provided.

## 1. Introduction

The Empididae (Diptera, Empidoidea) is a family of primarily predaceous flies, for which the classification, although not yet definitively established, has evolved considerably over the last twenty years (Sinclair and Cumming (2006) [1], Cumming et al. (2014) [2] and (2016) [3], Sinclair and Daugeron (2017) [4], Wahlberg and Johanson (2018) [5], and Sinclair and Shamshev (2021) [6]). The subfamily Empidinae forms the most diversified group of empidid flies and includes about 2000 described species. It displays an extraordinary range of mating behaviors (e.g., see Kessel (1955) [7], Chvála (1976) [8], Svensson and Petersson (1987) [9], Cumming (1994) [10], Funk and Tallamy (2000) [11], Sadowski et al. (1999) [12], and Murray et al. (2022) [13] and plays an important role in pollination, particularly in temperate areas and mountain environments (Lefebvre et al. (2018) [14]), but perhaps also in tropical areas where some genera of the tribe Empidini are considerably diversified (e.g., see Rafael and Cumming (2004) [15]).

The Empidinae are traditionally divided into two tribes, the Empidini and Hilarini. The representatives of the widely distributed Hilarini are distinguished by a bare laterotergite, costa circumambient, wing vein R_1_ generally thickened before joining the costal vein, and male first fore tarsomere often, but not always, swollen (e.g., see Bickel (1996) [16], (2002) [17], and (2023) [18]; Sinclair and Cumming (2006) [1]). Among the tribe Hilarini, the nuptial gift often consists of dead insects scavenged from the water surface, where adults form large swarms or aggregations (Plant (2004) [19]), and gifts are often wrapped in silk produced by specialized cells on the male fore tarsus (Sutherland et al. (2007) [20] and Young and Merritt (2003) [21]). When the gift item is a prey, the females feed on it during mating, and therefore prey is an important source of protein for egg maturation, but sometimes the gift consists of an inedible object (such as a seed) or just an empty silk cocoon (Tréhen (1965) [22]).

The Hilarini includes about 600 species worldwide, divided into 19 genera (Sinclair and Cumming (2006) [1]; Bickel (1996) [16], (2002) [17], and (2006) [23,24]; Ding et al. (2020) [25]). Eight genera have less than 10 species (eight are monotypic), while *Hilara* Meigen, 1822 and *Hilarempis* Bezzi, 1905 are the richest genera with about 400 and 110 species, respectively. The genus *Hilara* is worldwide and seems to be much more diversified in the Northern Hemisphere, probably because this fauna has been much better studied. The remaining genera are all distributed in the Southern Hemisphere as follows: Seven genera are endemic to the Australasian Realm, seven genera are endemic to South America, one genus is endemic to Africa, and two genera (*Hilarempis* and *Atrichopleura* Bezzi, 1910) have a wide distribution and are present in the Neotropical, Afrotropical, and Australasian Realms (Yang et al. (2007) [26]). In South America, the tribe is well known to be diversified in temperate areas, particularly in Patagonia (Collin (1933) [27]). In contrast, the Brazilian fauna of Hilarini is composed of only 18 species in four genera (Rafael and Câmara (2023) [28]), and most of them were treated in context with keys by Smith (1962) [29]. After that, Rafael (2001) [30] described two more species of *Hilara* from Pico da Neblina, Amazonas state. The last described Hilarini species for Brazil was *Atrichopleura acuminata* Câmara, Limeira-de-Oliveira and Rafael, 2013 (Câmara et al. (2013) [31]). Many Neotropical species are awaiting revisionary work and descriptions.

The purpose of this paper is to describe a distinctive new hilarine genus, namely *Tropicohilara* **gen. nov.**, as well as its six new species, i.e., one species from the Amazon rainforest and five species from the Atlantic Forest biome.

## 2. Materials and Methods

This study was based on specimens that will be deposited in the following institutions: the Invertebrate Collection of the Instituto Nacional de Pesquisas da Amazônia (INPA), Manaus, Amazonas, Brazil; the Museu Nacional do Rio de Janeiro (MNRJ), Rio de Janeiro, Brazil; the Museu de Zoologia da Universidade de São Paulo (MZUSP), São Paulo, Brazil, and the Muséum national d’Histoire naturelle (MNHN), Paris, France. The specimens were collected with Malaise traps, Shannon traps, and light traps in the Amazonian biome of Amazonas state, and the Atlantic Forest biomes of Bahia, Minas Gerais, Paraná, and Pernambuco states. The registration number in the National System for the Management of Genetic Heritage and Associated Traditional Knowledge (SisGen) is AF8BE54.

The general morphological terminology follows Cumming and Wood (2017) [32]. The specimen length was measured in lateral view from the frons (excluding antenna) to the apex of the abdomen.

The male genitalia of the specimens were macerated in hot (150 °C) 85% lactic acid for approximately 30 min, and then examined in glycerin on an excavated slide. After study and illustration, the dissected terminalia were placed in a microvial with glycerin, and then mounted below the specimen on the same pin.

The photographs were taken using a Leica DFC500 digital camera fitted onto a Leica MZ205 stereomicroscope connected to a computer with the Leica Application Suite LAS V3.6 software, including an Auto-Montage module (Syncroscopy software) which combines multiple serial layers of photographs into a single fully focused image.

Label data for the same specimen, but from different labels, are separated by quotation marks (“”). The original label data have been given verbatim for the type specimens. Square brackets ([ ]) are used to indicate complementary data not included on the labels.

## 3. Taxonomy

### 3.1. Tropicohilara ***gen. nov.***


urn:lsid:zoobank.org:act:589E8A8E-CDDE-4035-82B0-FC29CD0A2BCB


Type species: *Tropicohilara amazonensis* **sp. nov.**Gender: feminine

**Etymology**: The generic name is derived from Greek *tropikus*, i.e., a turning or solstice, and the generic name *Hilara*, and refers to the tropical distribution of the new genus.

**Diagnosis**: Body slender and predominantly yellow, occiput somewhat conical in dorsal view, postcranium yellow, with sparse grey pruinosity; frons and face shiny yellow without pruinosity; postpedicel elongate; scutum entirely yellow or with dark spots or stripes present; acrostichals, dorsocentrals, and supra-alars reduced to setulae; laterotergite bare; all legs slender; fore tibia without distinct anteroapical comb; hind tibia slightly shorter than hind femur; male first fore tarsomere not swollen, almost equal in length to the length of tarsomeres 2–5 combined; all legs densely covered mostly with short black setae, without strong setae; costal vein circumambient; vein R_1_ distinctly swollen before joining costa; vein Sc incomplete; vein R_1_ bare; vein R_2+3_ with setae on ventral surface; vein R_4+5_ branched, with R_4_ diverging at 45° angle and slightly sinuate, and R_5_ ending beyond wing apex; abdominal tergites covered with many distinct fine setae dorsally and laterally; male tergite 7 with a sclerotized band at posterior margin and tergite 8 reduced and upward directed so that terminalia can be flexed forward; male cercus divided into a smaller basal plate and cercus, this with an apical sinus and fused laterally with subepandrial sclerite; epandrium with distinct posterodorsal lobe; postgonite sinuous, widened medially; ejaculatory apodeme flattened, plate-like, dorsal lamella wider; phallus elongate and conforming to hypandrium curvature.

**Description**: Body length 4.6–6.6 mm. *Head* (Figure 1A–C,3A–D, 6A–D, 8A–D, 10A–D and 13A–D): Occiput yellow, somewhat conical in dorsal view (e.g., Figure 1C); postcranium with pale yellow or black setae, postocular setae short; postvertical setae very short, black; ocellar triangle distinct with reduced setae; frons shiny yellow usually slightly narrower than the ocellar triangle, with very short setae along lateral margins (e.g., Figure 1B); eyes slightly notched laterad of antennae; face wider than frons (e.g., Figure 1B), slightly expanded apically, shiny with very short setae along lateral margins; antenna with scape about twice as long as pedicel (e.g., Figure 1C); postpedicel elongate, tapering, longer than scape and pedicel combined, with short 2–articled apical stylus; palpus yellow with black setae, arched forward and slightly clavate; labrum tapering and well sclerotized, about 1.2 x head height in length, projecting ventrally. *Thorax* (Figure 1C, Figures 3D, 6D, 8D, 10D and 13D): Entirely yellowish-orange or with dark spots or stripes on mesoscutum, subshiny, without pruinosity; acrostichals and dorsocentrals reduced to setulae, former irregularly biserial, latter uniserial; 1–2 posterior notopleurals, 1 short postalar; one row of supra-alar setulae; antepronotum with some short setae; proepisternum covered with pale setae; postpronotal lobe with setulae; posterior region of anepisternum with pale setae; mesopleuron usually with silvery pruinose areas; laterotergite bare; scutellum and mediotergite light yellow to yellow. *Legs* (Figure 1A, Figures 3A, 6A, 8A, 10A and 13A): Fore and mid legs usually entirely yellow, hind leg entirely yellow or with femur and tibia partially or entirely dark brown to black, setae black. All tarsi with stout black claws and large yellowish pulvilli; all legs slender, with short and dense setae, without stout setae; fore tibia without distinct anteroapical comb; male fore tarsomere 1 not swollen; hind femur with short white pubescence ventrally; hind tibia slightly shorter than hind femur; hind tibia and hind tarsomere 1 with distinct posteroapical comb. *Wing* (Figure 1D, Figures 3A, 5A, 6A, 8E, 10A and 12A): Membrane hyaline or slightly yellow/light brown infuscate, sometimes dark brown on basal costal and costal cells; costal vein circumambient, reduced in thickness along posterior margin; Sc distinctly incomplete; vein R_1_ slightly swollen before joining costal vein; pterostigma brown, present under distal R_1_ (not visible in *T. amazonensis* **sp. nov.**, *T. bahiensis* **sp. nov.**, and *T. bella* **sp. nov.**); vein R_2+3_ haired on ventral surface (Figure 1D); vein R_4+5_ branched, with R_4_ diverging at 45°, slightly sinuate, vein R_5_ ending slightly beyond wing apex; veins M_1_, M_2_, and M_4_ reaching wing margin; lower calypter yellowish with yellow setae. *Abdomen* (Figure 1E, Figures 3E, 6E, 8F, 10E and 13E): Tergites 1–6 with little pruinosity and with some short fine setae dorsally and laterally, without stout marginal setae; tergite 1 relatively short; tergites 2–7 somewhat rectangular shaped; tergite 7 clearer, somewhat yellowish, with dark sclerotized band at posterior margin (e.g., Figure 2B and Figure 4B); tergite 8 upwardly directed, reduced, smaller than sternite 8. Terminalia. Male cercus divided, with smaller basal cercal plate and cercus with apical sinus, fused laterally with subepandrial sclerite, both long setose; epandrium wide, with distinct curved posterodorsal lobe. Hypandrium keel-shaped, membranous basally; postgonite sinuous, expanded medially. Ejaculatory apodeme flattened, plate-like, dorsal lamella wider. Phallus elongate and conforming to hypandrium curvature. Female abdomen telescoping (e.g., Figure 5A,B); oviscapt relatively unmodified, with well-developed tergites and sternites of segments 8–10, cercus subtriangular, setulose; genital fork reduced to small rod slightly wider distally; spermatheca spherical (e.g., Figure 5E).

**Remarks**: *Tropicohilara* **gen. nov.** comprises six species from the Amazon and Atlantic Forest biomes. Species occur in rainforests, and probably at higher altitudes. The only species collected in the Amazon biome, i.e., *T. amazonensis* **sp. nov**., was collected with Malaise and light traps set on a metal tower which was level with the top of the canopy at 32 m in a tropical forest north of Manaus, Amazonas, indicating that this species also inhabits the canopy.

*Tropicohilara* **gen. nov.** is easily distinguished from other genera by the combination of the following characteristics: species predominantly yellow having the occiput somewhat conical in dorsal view, the postpedicel elongate, the male first fore tarsomere not swollen, a sclerotized dark band at the posterior margin of male tergite 7, and setae on the ventral surface of vein R_2+3_. This last characteristic is also found in the monotypic Chilean genus *Pasitrichotus* Collin, 1933, however setae are present on both the ventral and dorsal sides of the vein; in addition to setae on R_2+3_, most veins of *Pasitrichotus* also have setae, both beneath and above.

In Empidinae, setae on the wing veins are also observed on the dorsal side of vein R_1_ of some undescribed species of *Hilara* from Australia and some Patagonian species of the tribe Empidini (e.g., *Empis fulvicollis* Collin, 1933), and on the dorsal and ventral sides of vein R_1_ in species of the genus *Deuteragonista* Philippi, 1865 (Collin 1933) [27].

### 3.2. Key to Species of Tropicohilara ***gen. nov.***

Scutum yellow with dark spots or stripes (Figures 6D, 8D and 13B) ………..………..2
-Scutum entirely yellow (Figure 1C, Figure 3D and 10D) …………………………..……4Scutum with four paramedian dark spots (Figure 6D). Hind femur almost entirely light yellow with a small dark brown ring apically (Figure 6A)………***T*. *bella* sp. nov.**
-Scutum with one anterior spot (Figure 13B) or paired dark brown stripes (Figure 8D). Hind femur predominantly brown, yellow basally (Figures 8A and 13A) ……………………………………………………………………………………………………………………… 3Scutum with one short dark brown spot anteriorly (Figure 13B,D). Tergite 8 with wide sinus basally and narrower sinus distally (Figure 14B) …… ***T*. *sinclairi* sp. nov.**
-Scutum with two paired dark brown stripes fused anteriorly (Figure 8D). Tergite 8 somewhat straight basally, with narrow sinus distally (Figure 9B) ………………………………………………………………………….… ***T*. *mineira* sp. nov.**All femora entirely yellow (Figure 1A)………………………..…***T*. *amazonensis* sp. nov.**
-At least apical half of hind femora dark brown to black (Figure 3A and Figure 10A)…5Halter entirely yellow. Tergite 8 with wider sinus basally and narrower sinus distally (Figure 4B). Male postgonite relatively slender along its length and not bifurcated at tip (Figure 4F). …….……………………………………………………***T*. *bahiensis* sp. nov.**
-Halter dark brown to black (Figure 10E). Tergite 8 with anterior and posterior margins somewhat straight (Figure 11B). Male postgonite more robust and bifurcated at the tip (Figure 11F) …….……………………… ***T*. *paranaensis* sp. nov.**

### 3.3. Tropicohilara amazonensis ***sp. nov.***

(Figure 1A–E and Figure 2A–G)


urn:lsid:zoobank.org:act:093547CA-A0B1-4D88-B348-E0B4C04AE6E2


**Type material: Holotype** Male (pinned, dissected): “BRASIL: Amazonas, Manaus, ZF-2, Km-14, 2°35′21″ S, 60°06′55″ W, 13–31.iii.2018, Malaise g[ran]de, 32 m alt, poente, J.A. Rafael—Rede BIA” “DW HIL 32” (Figure 2H) (INPA). **Paratypes:** Same data as holotype except “1–16.iv.2018” (1 male, INPA); same data as holotype except “17–30.iv.2018, Malaise g[ran]de, 32 m alt, nascente”, “EMP 599” (1 male, MNHN); same data as holotype except “Torre, 16–19.iv.2004, luz mista/BLB, lençol” “40 [=32] mts altura, J.A. Rafael, C.S. Motta, A. Silva Fº, J.M.F. Ribeiro”, “DW HIL 33“ (1 male, INPA); same data as holotype except“18–21.v.2004, lençol, luz mista e BLB”, “40 [=32] mts altura, J.A. Rafael, F.B. Baccaro, F.F. Xavier Fº & A. Silva Fº.” (2 males, INPA); same data as holotype except “15–18.vi.2004”, “J.A. Rafael, C.S. Motta, F. Godoi, S. Trovisco & A. Silva” (1 male, MNRJ); same data as holotype except “Rod. AM 010, Km 50, ZF-2, Km-14, Torre, 3–4.iii.2011, 21–00:00 h, Arm. Luz dossel, 40 m [32 m] de altura, FF Xavier Filho, J.T. Câmara; P. Dias leg” (1 male, MZSP).

**Diagnosis**: Labrum yellow except tip dark brown to black. Scutum, all femora, and tibiae entirely yellow. Wing membrane slightly brown infuscated, pterostigma not visible. Tergite 1, base and posterior margin of tergite 2 and tergite 7 yellow, remaining tergites mostly brown dorsally, yellow on posterior and lateral margins; tergite 7 yellow with dark sclerotized band on posterior margin; terminalia yellow.

**Description: Male.** Length 5.7 mm. *Head* (Figure 1A–C): Ocellar triangle yellow with whitish ocelli. Postcranium yellow, with sparse grey pruinosity and short pale setae; labrum shiny yellow but dark brown to black at tip; labellum yellow; antennal scape, pedicel and base of postpedicel yellowish brown, remaining of postpedicel black. *Thorax* (Figure 1A,C) yellowish-orange with pale yellow postpronotal lobe, without distinct pruinosity; mesopleuron yellow with whitish areas ventrally; scutellum with three pairs of brownish marginal setae, some adjacent short setae. *Legs* (Figure 1A): All legs yellow except hind tarsus whitish. *Wing* (Figure 1A,D): Length 5.9 mm. Slightly brown infuscate; bc and c cells light brown. Halter whitish yellow. *Abdomen* (Figure 1A,E): Tergite 1, base and posterior margin of tergite 2 and tergite 7 yellow, remaining tergites mostly brown dorsally, yellow on posterior and lateral margins; tergite 7 light yellow with narrow dark sclerotized band on posterior margin. Tergites 2–7 densely covered with relatively short setae dorsally and laterally. Sternites yellow, covered with short pale setae (Figure 2A,C). Tergite 8 reduced, rectangular, with black setae on posterior margin, slightly lighter anteromedially (Figure 2A,B). Terminalia (Figure 2A–G) mainly yellow. Postgonite with same thickness along most of its length tapered apically, with small sinus at tip (Figure 2G). **Female.** Unknown

**Etymology:** The specific name refers to the state where the specimens were collected, Amazonas, Brazil.

**Geographical record:** Brazil (Amazonas state, Manaus) (Figure 16).

**Bionomics:** Specimens were collected only in the canopy with Malaise and light traps from March to June.

**Figure 1 insects-14-00915-f001:**
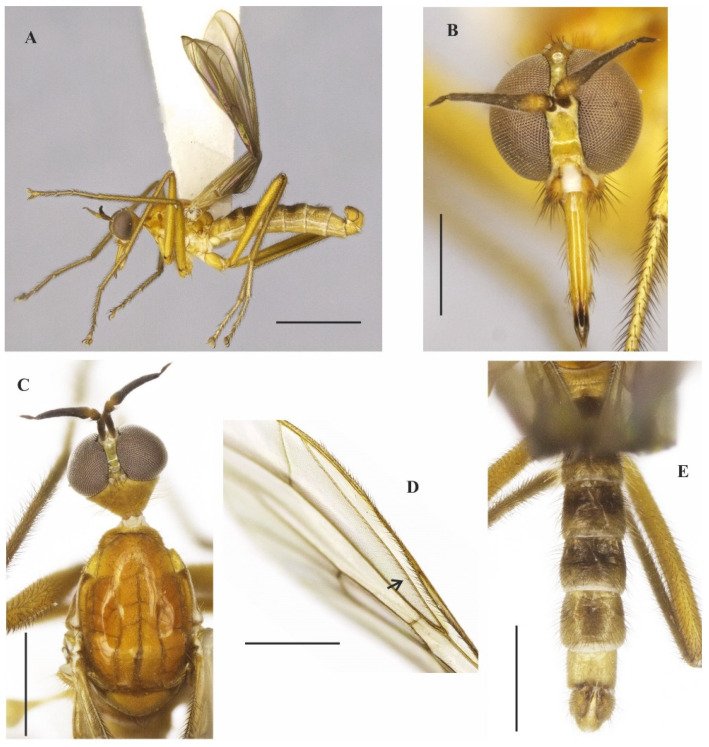
*Tropicohilara amazonensis* **sp. nov.**: (**A**) Habitus (holotype male), lateral view, scale bar = 2 mm; (**B**) head (holotype male), frontal view, scale bar = 0.5 mm; (**C**) head and thorax (holotype male), dorsal view, scale bar = 1 mm; (**D**) wing (paratype male), arrow indicating the setae on vein R_2+3_, scale bar = 1 mm; (**E**) abdomen (holotype male), dorsal view, scale bar = 1 mm.

**Figure 2 insects-14-00915-f002:**
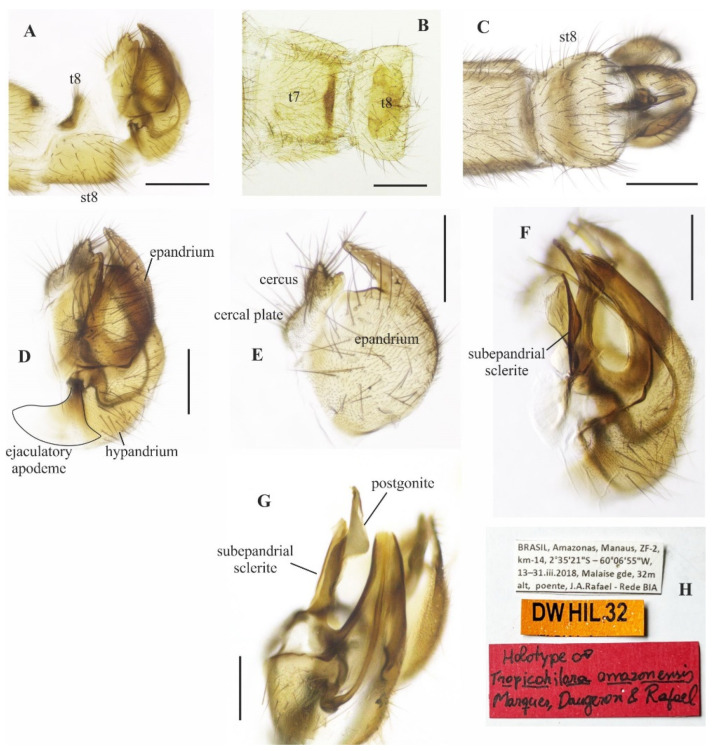
*Tropicohilara amazonensis* **sp. nov.**, holotype male: (**A**) Tergite and sternite 8, and terminalia, lateral view, scale bar = 0.3 mm; (**B**) tergite 7, tergite 8, dorsal view, scale bar = 0.3 mm; (**C**) sternite 8 and terminalia, ventral view, scale bar = 0.3 mm; (**D**) terminalia, lateral view, ejaculatory apodeme outlined, scale bar = 0.2 mm (**E**) epandrium and cercus, lateral view, scale bar = 0.2 mm; (**F**) terminalia with epandrium and cercus detached, lateral view, scale bar = 0.2 mm; (**G**) terminalia, detail of postgonite, posterolateral view, scale bar = 0.2 mm; (**H**) holotype labels.

### 3.4. Tropicohilara bahiensis ***sp. nov.***

(Figure 3A–E, Figure 4A–G and Figure 5A–F)


urn:lsid:zoobank.org:act:650A7194-17CC-4EAC-B73D-0E5E2240CA4A


**Type material**: **Holotype** male (pinned, terminalia in microvial of glass): “BRASIL, BA[Bahia], Camacan, Res.[erva] Serra Bonita, 152330 S—393357 W”, “820 m, 08–09.v.2007, J.A. Rafael & F.F. Xavier Fº, luz”, “DW HIL 39” (Figure 4G) (INPA). **Paratypes**: Same data as holotype (1 male, 1 female, INPA).

**Diagnosis:** Labrum reddish brown basally, grading to black on extreme apex. Scutum entirely yellowish orange. Fore and mid legs dark yellow; hind leg yellow except apical two thirds of femur dark brown and basal half of tibia brownish. Wing membrane slightly brown infuscate, pterostigma not visible. Abdominal tergites 1–6 dark brown to black, tergite 7 dark yellow with a dark sclerotized band on posterior margin. Terminalia yellowish.

**Description: Male**. Length 5.6 mm. *Head* (Figure 3A–D): Ocellar triangle orange yellow with yellowish ocelli. Postcranium with pale setae; labrum reddish brown basally, grading to black on apical third; labellum yellow; antennal scape and pedicel brownish yellow, postpedicel dark brown to black. *Thorax* (Figure 3D): Entirely yellowish orange with pale yellow postpronotal lobe and postalar callus, without distinct pruinosity; scutellum with fringe of brownish marginal setae. *Legs* (Figure 3A): Fore and mid legs dark yellow; hind leg yellow except apical two thirds of femur dark brown and basal half of tibia brownish. *Wing*: Length 6.3 mm. Slightly light brown infuscate, pterostigma not visible. Halter yellow. *Abdomen* (Figure 3E): Tergites 1–6 dark brown to black, tergite 7 dark yellow with dark sclerotized band on posterior margin (Figure 4A,B). Tergites 2–7 covered with relatively short setae dorsally, longer setae laterally. Tergite 8 reduced, with wider sinus on anterior margin and narrower sinus on posterior margin, with black setae on posterior margin (Figure 4B). Terminalia (Figure 4A–F) yellowish. Postgonite widened medially, rounded at tip (Figure 4F). **Female** (Figure 5A–F). 6.4 mm. Similar to the male, thorax and abdomen with similar color pattern (Figure 5A). Terminalia. Tergite 8 concave on anterior margin, with distinct black setae on posterior margin (Figure 5B,C); tergite 9+10 somewhat pentagonal (Figure 5C); sternite 8 with posterior margin sinuous (Figure 5D); sternite 10 with bare area medially (Figure 5D); cercus well developed, triangular (Figure 5B–D); spermatheca spherical (Figure 5E). Egg slightly oval (Figure 5F). 

**Etymology***:* The specific name refers to the state where the specimens were collected, Bahia, Brazil.

**Geographical record:** Brazil (Bahia state) (Figure 16). 

**Bionomics**: Specimens were collected on May using a light trap.

**Figure 3 insects-14-00915-f003:**
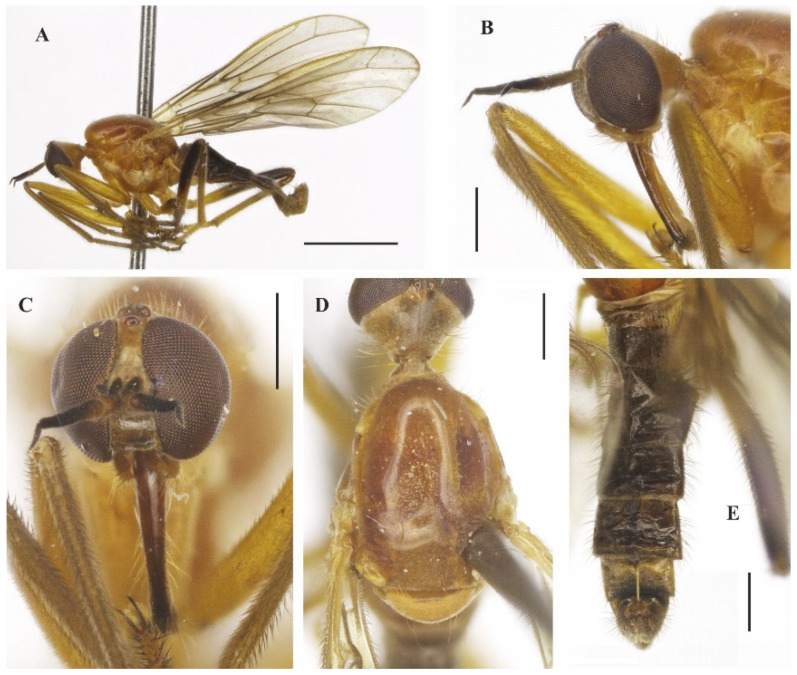
*Tropicohilara bahiensis* **sp. nov.**, holotype male: (**A**) Habitus, lateral view, scale bar = 2 mm; (**B**) head, lateral view, scale bar = 0.5 mm; (**C**) head, frontal view, scale bar = 0.5 mm; (**D**) thorax, dorsal view, scale bar = 0.5 mm; (**E**) abdomen, dorsal view, scale bar = 0.5 mm.

**Figure 4 insects-14-00915-f004:**
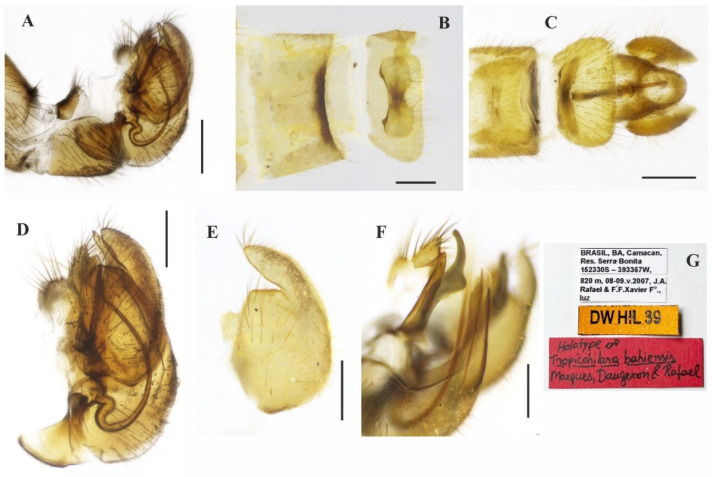
*Tropicohilara bahiensis* **sp. nov.**, holotype male: (**A**) Tergite and sternite 8, and terminalia, lateral view, scale bar = 0.3 mm; (**B**) posterior margin of tergite 7 and tergite 8, dorsal view, scale bar = 0.3 mm; (**C**) sternite 8 and terminalia, ventral view, scale bar = 0.3 mm; (**D**) terminalia, lateral view, scale bar = 0.2 mm; (**E**) epandrium detached, lateral view, scale bar = 0.2 mm; (**F**) terminalia, detail of postgonite, posterolateral view, scale bar = 0.2 mm; (**G**) holotype labels.

**Figure 5 insects-14-00915-f005:**
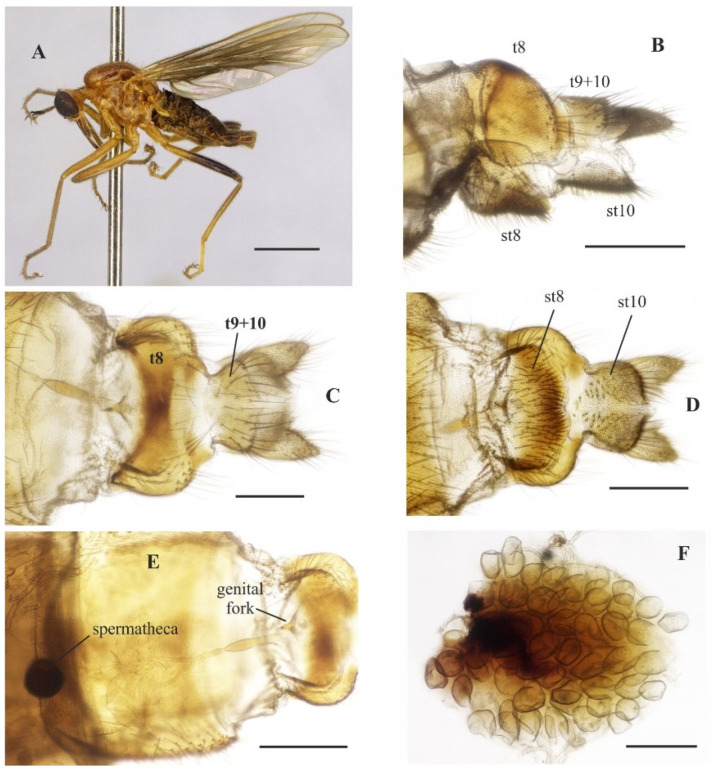
*Tropicohilara bahiensis* **sp. nov.**, paratype female: (**A**) Habitus, lateral view, scale bar = 2 mm; (**B**) apex of abdomen, lateral view, scale bar = 0.3 mm; (**C**) idem, dorsal view, scale bar = 0.2 mm; (**D**) idem, ventral view, scale bar = 0.3 mm; (**E**) genital fork and spermatheca, scale bar = 0.3 mm; (**F**) eggs, scale bar = 0.5 mm.

### 3.5. Tropicohilara bella ***sp. nov.***

(Figure 6A–E and Figure 7A–G)


urn:lsid:zoobank.org:act:C12577D9-86D0-47EE-B76C-4FF0C1841866


**Type material**: **Holotype** male (pinned, terminalia in microvial of glass): “BRASIL, PE[Pernambuco], Jaqueira, RPPN Frei Caneca, 084315 S—355027 W”, “600 m, 28.v.2007, J.A. Rafael & F.F. Xavier Fº., luz”, “DW HIL 35” (Figure 7G, INPA).

**Diagnosis:** Labrum yellow except tip black. Scutum yellow with two pairs of large black spots laterally, between the dorsocentral rows and notopleuron. All legs yellow except hind femora with dark brown ring apically, hind tarsomeres whitish yellow. Wing membrane hyaline, pterostigma not visible. Abdominal tergites 1–6 and 8 mostly dark brown dorsally, yellowish on posterior and lateral margins; tergite 7 yellow with a dark sclerotized band on posterior margin; terminalia brown. 

**Description: Male.** Length 4.6 mm. *Head* (Figure 6A–D): Ocellar triangle dark brown to black with yellowish ocelli. Postcranium with short yellow setae. Labrum yellow, except tip black; labellum yellow; antennal scape and pedicel yellowish brown, postpedicel yellow at base remaining dark brown to black. *Thorax* (Figure 6D): Yellow, with two pairs of large black spots laterally on mesoscutum, between the dorsocentral rows and notopleuron; scutellum with fringe of brownish marginal setae. *Legs* (Figure 6A): All legs yellow except hind femora with dark brown ring apically, hind tarsomeres whitish yellow. *Wing*: Length 6.1 mm. Membrane hyaline, pterostigma not visible; bc and c cells light brown infuscate. Halter yellow. *Abdomen* (Figure 6E); Tergites 1–6 mostly dark brown dorsally, yellowish on posterior and lateral margins; tergite 7 yellow with dark sclerotized band on posterior margin (Figure 7A,B). Tergites densely covered with relatively short setae dorsally and laterally. Tergite 8 reduced, brown, with small medial sinus and black setae on posterior margin (Figure 7A,B). Sternites 1–7 yellow, covered with short pale setae; sternite 8 brown (Figure 7A,C). Terminalia with dark brown epandrium and hypandrium, but distinctly yellow cercus (Figure 7A,C–F); epandrium clearer emarginate on dorsal margin (Figure 7E). Postgonite robust, widened medially (Figure 7F). **Female.** Unknown.

**Etymology:** The specific name derives from the Latin word *bellus* = beautiful.

**Geographical record:** Brazil (Pernambuco state) (Figure 16).

**Bionomics:** The single known specimen of this species was collected in May using a light trap.

**Figure 6 insects-14-00915-f006:**
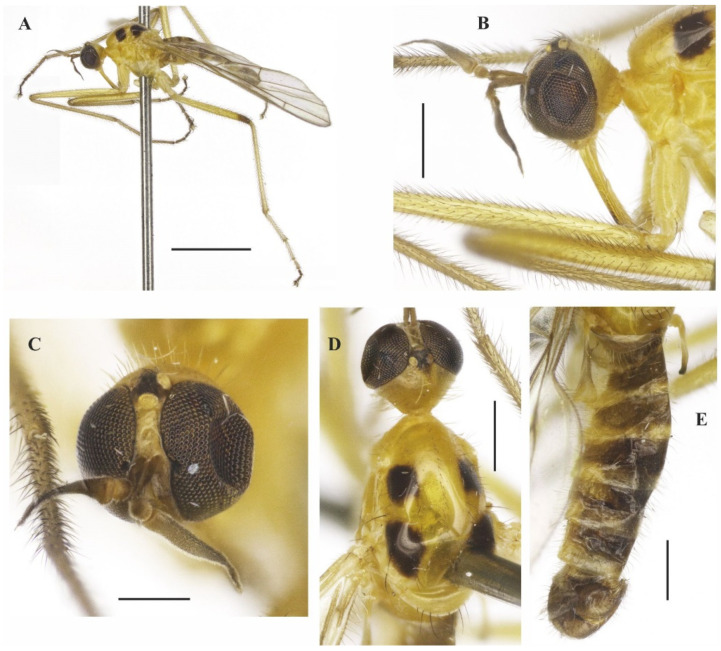
*Tropicohilara bella* **sp. nov.**, holotype male: (**A**) Habitus, lateral view, scale bar = 2 mm; (**B**) head, lateral view, scale bar = 0.5 mm; (**C**) head, frontal view, scale bar = 0.3 mm; (**D**) head and thorax, dorsal view, scale bar = 0.5 mm; (**E**) abdomen, dorsal view, scale bar = 0.5 mm.

**Figure 7 insects-14-00915-f007:**
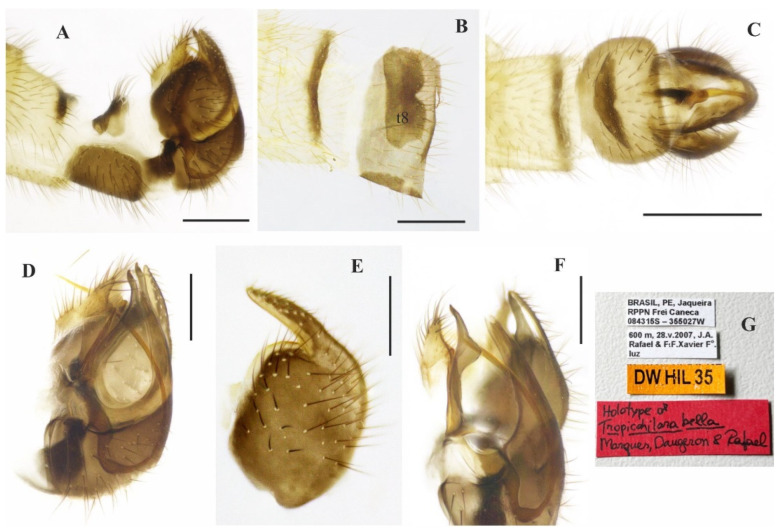
*Tropicohilara bella* **sp. nov.**, holotype male: (**A**) Tergite and sternite 8, and terminalia, lateral view, scale bar = 0.3 mm; (**B**) tergite 7 and tergite 8, dorsal view, scale bar = 0.3 mm; (**C**) sternites 7 and 8, and terminalia, ventral view, scale bar = 0.3 mm; (**D**) terminalia, epandrium detached, lateral view, scale bar = 0.2 mm; (**E**) epandrium detached, lateral view, scale bar = 0.2 mm; (**F**) terminalia, detail of postgonite, posterolateral view, scale bar = 0.2 mm; (**G**) holotype labels.

### 3.6. Tropicohilara mineira ***sp. nov.***

(Figure 8A–F and Figure 9A–G)


urn:lsid:zoobank.org:act:062AF8B6-E78B-4CEB-AE4E-F8561E9480B7


**Type material**: **Holotype** male (pinned, terminalia in microvial of glass): “BRASIL, MG[Minas Gerais], Itamonte—Pq.[Parque] Nac.[ional] do Itatiaia—Brejo da Lapa—Capão de Araucárias, 22°21’23” S, 44°44’02” W”, “Shannon trap—2172 m, 7–14.Jan[uary].2016, [D.S.] Amorim & [V.C.] Silva cols”, “DW HIL 36” (Figure 9G, INPA). **Paratype:** Same data except, “Malaise trap” (1 male, MZUSP).

**Diagnosis:** Labrum reddish brown basally, grading to black on distal third. Scutum mostly yellow with two large reddish-brown stripes medially, between the acrostichal and dorsocentral setae. Fore and mid legs dark yellow; hind leg yellow except apical two thirds of femur dark brown and tibia yellowish brown. Wing membrane light brown infuscate except apex of basal costal and costal cells brown, pterostigma brown present under apex of vein R_1_. Abdominal tergites 1–6 and 8 mostly dark brown dorsally, whitish yellow on posterior and lateral margins, tergite 7 yellowish brown with dark sclerotized band on posterior margin. 

**Description: Male.** Length 5.2 mm. *Head* (Figure 8A–D): Ocellar triangle dark brown to black with yellowish ocelli. Postcranium with brownish setae. Labrum reddish brown basally, grading to black in apical third; palpus dark yellow; labellum yellow; antenna dark brown. *Thorax* (Figure 8D): Mostly yellow, mesoscutum with two large dark reddish-brown stripes medially, between the acrostichal and dorsocentral setae. Scutellum with two distinct pairs of brownish marginal setae, with some adjacent short setae. *Legs* (Figure 8A): Fore and mid legs dark yellow; hind leg yellow except apical two thirds of femur dark brown and tibia yellowish brown. *Wing* (Figure 8E): Length: 7.2 mm. Wing membrane light brown infuscate except basal costal cell and almost entire costal cell brown; pterostigma brown, narrow under apex of vein R_1_. Halter with base whitish yellow, remaining brown. *Abdomen* (Figure 8F): Tergites 1–6 and 8 mostly dark brown dorsally, whitish yellow on posterior and lateral margins; tergite 7 yellowish brown with dark sclerotized band on posterior margin (Figure 9B). Tergites 2–7 covered with relatively short pale setae dorsally and laterally. Tergite 8 reduced, brownish, darker posteriorly, somewhat rectangular, somewhat straight anteriorly and with narrow sinus on posterior margin (Figure 9B). Terminalia (Figure 9A,C–F) with dark brown epandrium, hypandrium, and cercus. Postgonite bifurcated at tip (Figure 9F). **Female.** Unknown.

**Etymology***:* Noun in apposition, refers to the state where the material of the new species was collected, Minas Gerais, Brazil.

**Geographical record:** Brazil (Minas Gerais state) (Figure 16).

**Bionomics:** Specimens were collected in January using Shannon and Malaise traps.

**Figure 8 insects-14-00915-f008:**
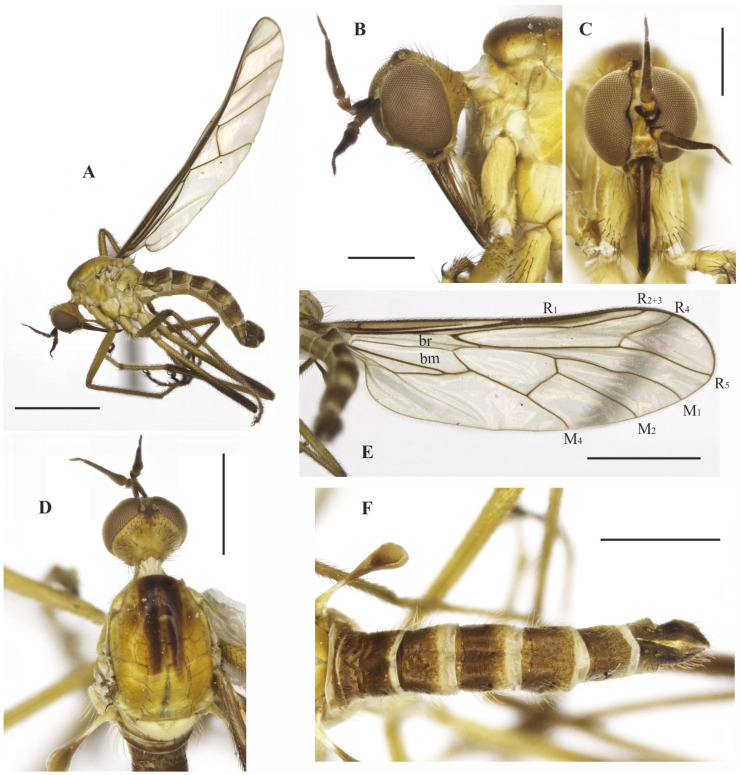
*Tropicohilara mineira* **sp. nov.**, holotype male: (**A**) Habitus, lateral view, scale bar = 2 mm; (**B**) head, lateral view, scale bar = 0.5 mm; (**C**) head, frontal view, scale bar = 0.5 mm; (**D**) head and thorax, dorsal view, scale bar = 1 mm; (**E**) wing, dorsal view, scale bar = 2 mm; (**F**) abdomen, dorsal view, scale bar = 2 mm.

**Figure 9 insects-14-00915-f009:**
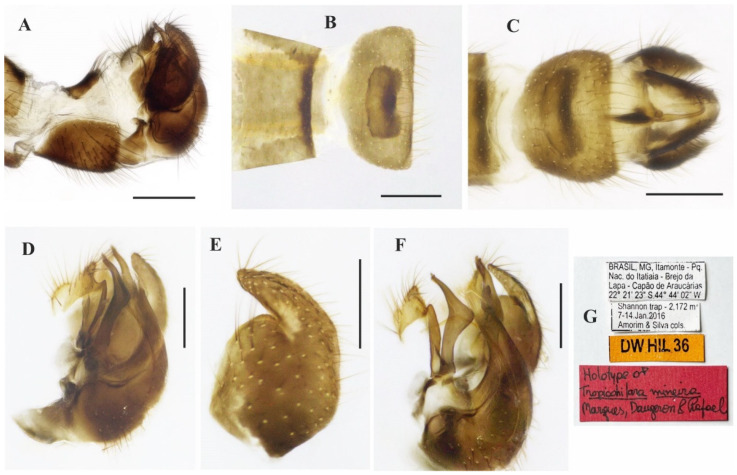
*Tropicohilara mineira* **sp. nov.**, holotype male: (**A**) Tergite and sternite 8, and terminalia, lateral view, scale bar = 0.3 mm; (**B**) tergites 7 and 8, dorsal view, scale bar = 0.3 mm; (**C**) sternite 8 and terminalia, ventral view, scale bar = 0.3 mm; (**D**) terminalia, epandrium detached, lateral view, scale bar = 0.2 mm; (**E**) epandrium detached, lateral view, scale bar = 0.2 mm; (**F**) terminalia, detail of postgonite, posterolateral view, scale bar = 0.2 mm; (**G**) holotype labels.

### 3.7. Tropicohilara paranaensis ***sp. nov.***

(Figure 10A–E, Figure 11A–G and Figure 12A–E)


urn:lsid:zoobank.org:act:8430C954-A8D1-4BC9-BAE4-AA9797303AE1


**Type material**: **Holotype** male (pinned, terminalia in microvial of glass): “BRASIL, PR[Paraná], Piraquara, Manancial da Serra Mar, 252946 S—485854 W”, “1000 m, 16–17.xii.2006, J.A. Rafael & G.A.R. Melo, Arm. Luminosa (lençol)”, “DW HIL 38” (Figure 11G, INPA). **Paratypes:** Same data as holotype (2 females, INPA). 

**Diagnosis** Labrum dark brown to black. Scutum entirely yellowish orange. Fore and mid legs yellow with tibiae and tarsomeres darker; hind leg yellow except distal two thirds of femur, entire tibia, and tarsomeres 4 and 5 dark brown to black. Wing membrane light brown infuscate except apex of basal costal and entire costal cells brown, pterostigma brown, narrow, under apex of vein R_1_. Abdominal tergite 1 with base yellow and remaining dark brown dorsally, tergites 2–6 and 8 mostly dark brown to black dorsally, somewhat yellowish laterally, tergite 7 yellow with a dark sclerotized band on posterior margin; sternites 1–7 yellow, sternite 8 brown. 

**Description: Male.** Length 5.3 mm. *Head* (Figure 10A–D): Ocellar triangle dark brown to black with yellowish ocelli. Postcranium with dark setae dorsally and pale setae ventrally. Labrum entirely dark brown to black; palpus dark yellow, brown at tip; labellum brownish; antenna dark brown to black. *Thorax* (Figure 10D): Entirely yellowish orange, without distinct pruinosity; scutellum with one stouter pair of median marginal setae, with some adjacent short setae. *Legs* (Figure 10A): Fore and mid legs yellow, tibiae and tarsomeres darker; hind leg yellow except apical two thirds of femur, entire tibia and tarsomeres 4 and 5 dark brown to black; all legs with coxae and trochanters with yellow setae. *Wing* (Figure 10A): Length 7.1 mm. Membrane slightly brown infuscate except apex of basal costal and entire costal cells brown; pterostigma brown under apex of vein R_1_. Halter with base yellow remaining dark brown to black. *Abdomen* (Figure 10D): Tergite 1 with base yellow and remaining dark brown dorsally, tergites 2–6 and 8 mostly dark brown to black dorsally, somewhat yellowish laterally, tergite 7 (Figure 11A,B) yellow with dark sclerotized band on posterior margin; sternites 1–7 yellow, sternite 8 brown (Figure 11A,C). Tergites densely covered with relatively short setae dorsally and laterally. Tergite 8 reduced, with black setae on posterior margin, with anterior and posterior margins somewhat straight (Figure 11A,B). Terminalia (Figure 11A,C–F) with dark brown epandrium, hypandrium, and cercus. Postgonite bifurcated at tip (Figure 11F). **Female.** Body length 6.6 mm. Similar to the male, thorax and abdomen with similar color pattern (Figure 12A). Terminalia (Figure 12B–E). Tergite 8 with wide sinus on anterior margin, smaller sinus medially and distinct black setae on posterior margin; posterior margin of tergite 9 + 10 and sternite 10 rounded and pointed, respectively; cercus triangular; spermatheca subspherical.

**Etymology:** The specific name refers to the state where the specimens were collected, Paraná, Brazil.

**Geographical record:** Brazil (Paraná state) (Figure 16).

**Bionomics:** Specimens were collected in December using a light trap.

**Figure 10 insects-14-00915-f010:**
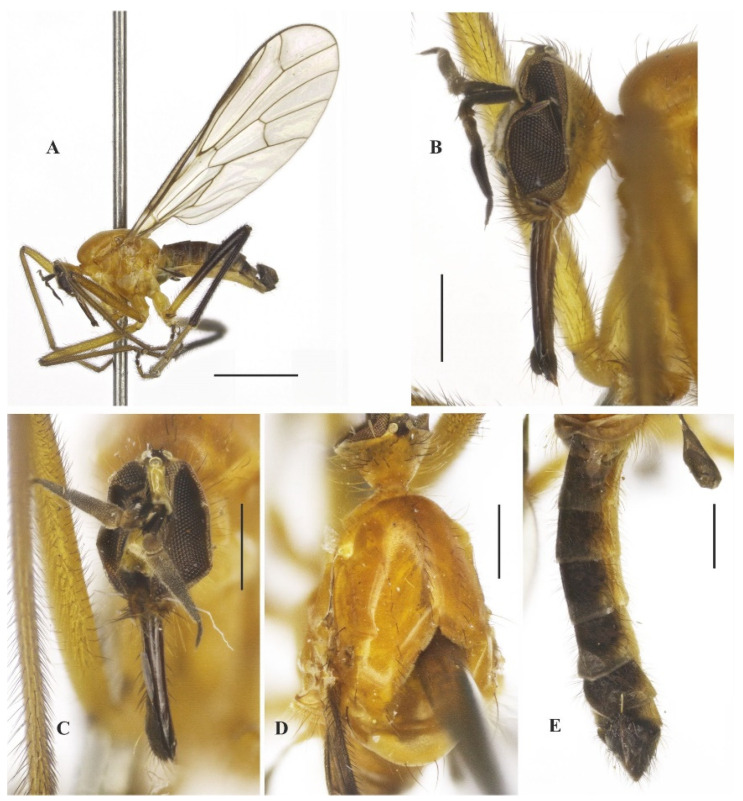
*Tropicohilara paranaensis* **sp. nov.**, holotype male: (**A**) Habitus, lateral view, scale bar = 2 mm; (**B**) head, lateral view, scale bar = 0.5 mm; (**C**) head, frontal view, scale bar = 0.5 mm; (**D**) head and thorax, dorsal view, scale bar = 0.5 mm; (**E**) abdomen, dorsal view, scale bar = 0.5 mm.

**Figure 11 insects-14-00915-f011:**
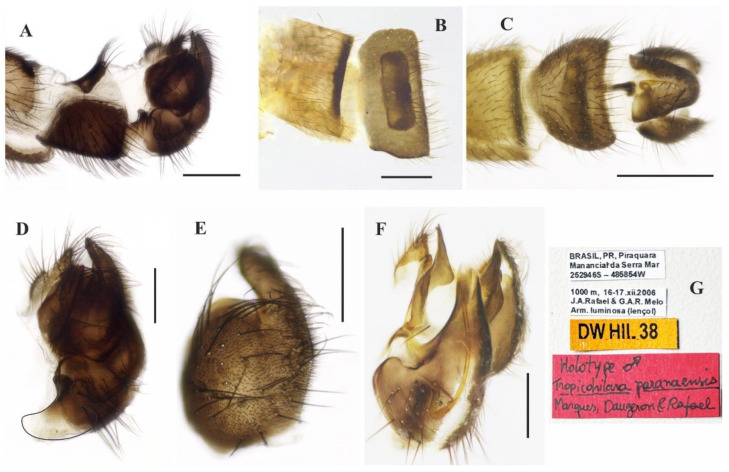
*Tropicohilara paranaensis* **sp. nov.**, holotype male: (**A**) Tergite and sternite 8, and terminalia, lateral view, scale bar = 0.3 mm; (**B**) tergites 7 and 8, dorsal view, scale bar = 0.3 mm; (**C**) sternite 8 and terminalia, ventral view, scale bar = 0.5 mm; (**D**) terminalia, lateral view, scale bar = 0.2 mm; (**E**) epandrium detached, lateral view, scale bar = 0.2 mm; (**F**) terminalia, detail of postgonite, posterolateral view, scale bar = 0.2 mm; (**G**) holotype labels, scale bar = 0.2 mm.

**Figure 12 insects-14-00915-f012:**
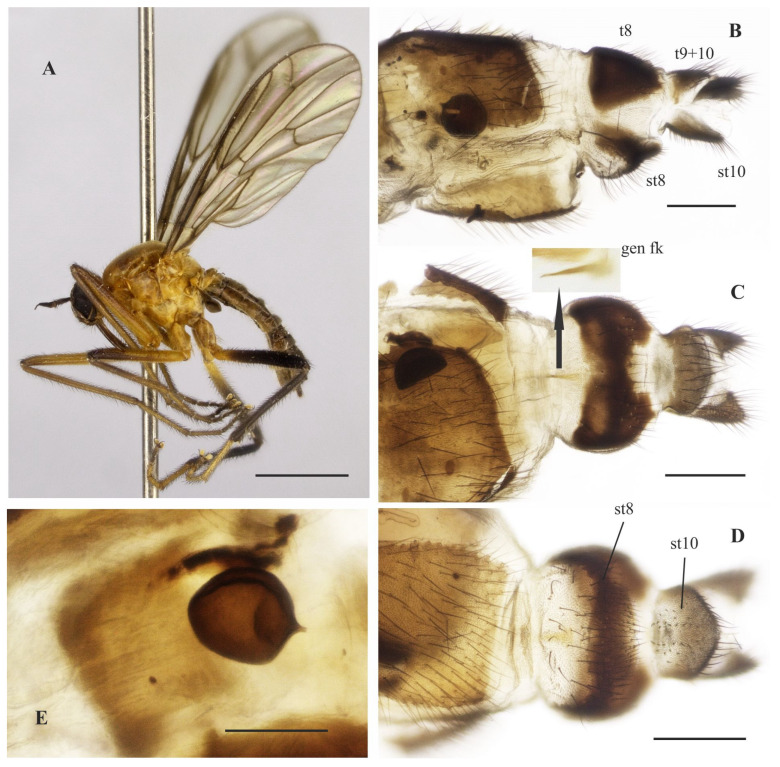
*Tropicohilara paranaensis* **sp. nov.**, paratype female: (**A**) Habitus, lateral view, scale bar = 2 mm; (**B**) apex of abdomen, lateral view, scale bar = 0.3 mm; (**C**) *idem*, dorsal view, scale bar = 0.3 mm; (**D**) *idem*, ventral view, scale bar = 0.3 mm; (**E**) spermatheca, scale bar = 0.2 mm.

### 3.8. Tropicohilara sinclairi ***sp. nov.***

(Figure 13A–E, Figure 14A–G and Figure 15A–F)


urn:lsid:zoobank.org:act:9A924734-D799-467F-8497-A2187045EBC0


**Type material**: **Holotype** male (pinned, terminalia in microvial of glass): “BRASIL, Paraná, Morretes, luz—CIIF [Centro de Identificação de Insetos Fitófagos], ix–x.1984” [Renato Marinoni col.], “DW HIL 34” (Figure 14G, INPA). **Paratypes:** Same data as holotype (3 males, 3 females, INPA; 2 males, 2 females, MNRJ); Morretes, 7. ix.1984, J.A.Rafael (1 female, INPA); “MG [Minas Gerais], Perdizes, EPDA Galheiro córrego-casa 4, 19°12′34″ S, 47°8′21″ W, Malaise trap, 15–29.i.2020, Capellari, Riccardi & Pirani leg”, “EMP 600”(1 female, MNHN). 

**Diagnosis:** Labrum yellow basally, grading to dark brown in apical third. Scutum mostly yellow with thin brown stripes extending over dorsocentral and acrostichal rows, and a dark brown spot anteriorly at level of acrostichal row. Fore and mid legs yellow; hind leg yellow except tibia and basal half of femur dark brown, hind tarsomeres whitish. Wing membrane hyaline, pterostigma light yellow. Abdominal tergites 1–6 and terminalia mostly dark brown dorsally, somewhat yellowish laterally, tergite 7 yellow with a dark sclerotized band on posterior margin.

**Description: Male.** Length: 5.8 mm. *Head* (Figure 13A–D): Ocellar triangle dark brown to black with reddish ocelli. Postcranium with usual pale setae; face with a shiny brown spot on the ventral margin. Labrum yellow basally, grading to dark brown on apical third; labellum yellow; antennal scape yellowish brown, scape yellow, and postpedicel yellow at base, remaining dark brown to black. *Thorax* (Figure 13D): Yellow with pale yellow postpronotal lobe and postalar callus, without distinct pruinosity; mesoscutum with thin brown stripes extending over dorsocentral and acrostichal rows, and dark brown spot anteriorly at level of acrostichal row (more distinct in paratypes); scutellum with fringe of brownish marginal setae. *Legs* (Figure 13A): Fore and mid legs yellow; hind leg yellow, except apical half of femur and entire tibia, dark brown, hind tarsomeres whitish. *Wing*: Length 6.4 mm. Membrane light brown infuscate, pterostigma whitish. Halter brown. *Abdomen* (Figure 13E and Figure 14B,C): Abdominal tergites 1–6 and 8 mostly dark brown dorsally, somewhat yellowish laterally, tergite 7 yellow with dark sclerotized band on posterior margin (Figure 14A,B). Tergites 2–7 densely covered with relatively short setae laterally. Tergite 8 reduced, somewhat rectangular, with black setae on posterior margin, with small medial sinus on anterior and posterior margins (Figure 14B). Sternites 1–7 yellow, covered with short pale setae; sternite 8 brown (Figure 14A,C). Terminalia (Figure 14A,C–F). Epandrium and hypandrium dark brown, cercus yellow. Postgonite (Figure 14F) robust, widened medially, bifurcated at tip. **Female** (Figure 15A–E). Body length 6.2 mm. Similar to the male, thorax and abdomen with similar color pattern (Figure 15A). Terminalia (Figure 15B–F). Tergite 8 longer laterally, narrower in middle, with black setae on posterior margin; tergite 9+10 with lateral margins somewhat rounded; sternite 10 with posterior margin pointed (Figure 15E); cercus subtriangular; spermatheca spherical. 

**Etymology***:* The species epithet is in honor of the entomologist Bradley Sinclair for his contribution to the authors and knowledge of the empidids.

**Geographical record:** Brazil (Paraná state) (Figure 16).

**Bionomics:** Specimens were collected in January, September, and October using Malaise and light traps.

**Figure 13 insects-14-00915-f013:**
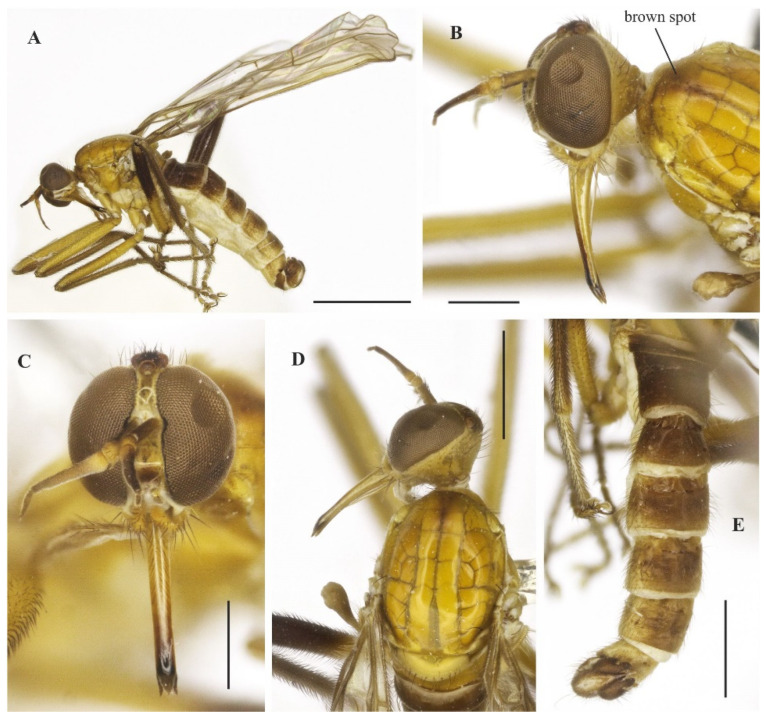
*Tropicohilara sinclairi* **sp. nov.**, holotype male: (**A**) Habitus, lateral view, scale bar = 2 mm; (**B**) head, lateral view, scale bar = 0.5 mm; (**C**) head, frontal view, scale bar = 0.5 mm; (**D**) head and thorax, dorsal view, scale bar = 1 mm; (**E**) abdomen, dorsal view, scale bar = 1 mm.

**Figure 14 insects-14-00915-f014:**
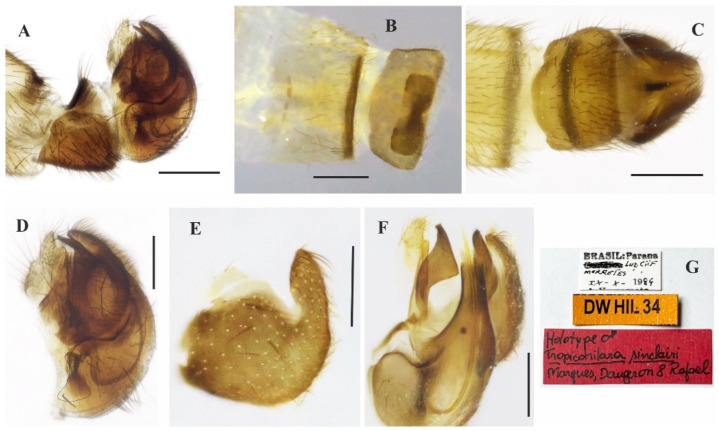
*Tropicohilara sinclairi* **sp. nov.**, holotype male: (**A**) Tergite and sternite 8, and terminalia, lateral view, scale bar = 0.3 mm; (**B**) tergites 7 and 8, dorsal view, scale bar = 0.3 mm; (**C**) sternite 8 and terminalia, ventral view, scale bar = 0.3 mm; (**D**) terminalia, ejaculatory apodeme outlined, lateral view, scale bar = 0.2 mm; (**E**) epandrium detached, lateral view, scale bar = 0.2 mm; (**F**) terminalia, detail of postgonite, posterolateral view, scale bar = 0.2 mm; (**G**) holotype labels.

**Figure 15 insects-14-00915-f015:**
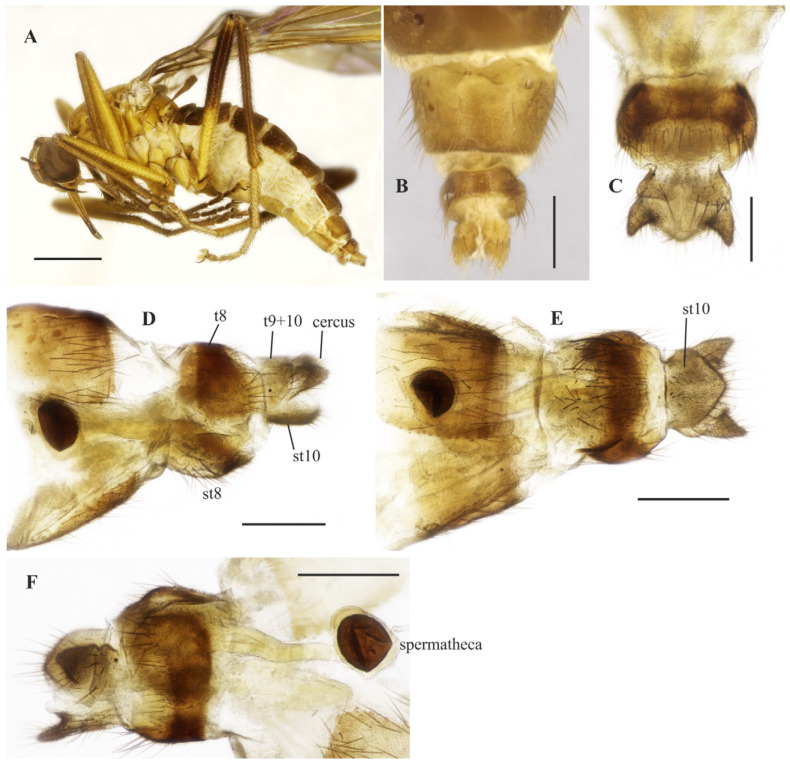
*Tropicohilara sinclairi* **sp. nov.**, paratype female: (**A**) habitus, lateral view, scale bar = 1 mm; (**B**) apex of abdomen, dorsal view, scale bar = 0.3 mm; (**C**) *idem*, dorsal view, scale bar = 0.2 mm; (**D**) *idem*, lateral view, scale bar = 0.3 mm; (**E**) *idem*, ventral view, scale bar = 0.3 mm; (**F**) spermatheca, dorsolateral view, scale bar = 0.3 mm.

**Figure 16 insects-14-00915-f016:**
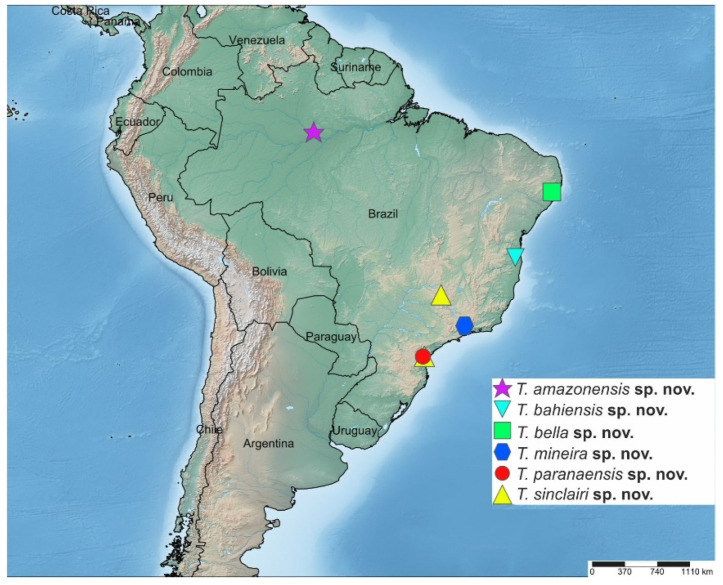
Map of South America showing the distribution of the six species of the genus *Tropicohilara* in Brazil.

## 4. Discussion

*Tropicohilara* **gen. nov.** comprises six distinct Brazilian species, with only one species occurring in the Amazon rainforest, while the remaining five species are concentrated within the Atlantic Forest biome. The species were collected in different Brazilian states, including Amazonas, Bahia, Pernambuco, Minas Gerais, and Paraná, showcasing the extensive range of this new genus across four of Brazil’s five geographic regions. An interesting facet of this distribution is that most of these species have been documented in the Atlantic Forest, a highly threatened biome, predominantly in high-altitude and mountainous regions. Notably, the sole species found in the Amazon, i.e., *T. amazonensis* **sp. nov.**, was collected using Malaise and light traps set on a metal tower that was level with the top of the canopy at 32 m in a tropical forest north of Manaus, Amazonas, suggesting this species inhabits the canopy. The collection method for the Amazonian species, involving canopy traps on a 32 m tower, highlights the importance of innovative and non-invasive research techniques in studying and conserving high-canopy rainforest species. 

*Tropicohilara* **gen. nov.** is characterized by an occiput that appears somewhat triangular when viewed dorsally (e.g., Figure 1C) and more or less horizontal when viewed laterally (e.g., Figure 3B), considered a potential synapomorphy for the genus. Another possible synapomorphy is the presence of a highly sclerotized band on the posterior margin of tergite 7, a characteristic that is present in all species. The shape of tergite 7 is sometimes modified in species belonging to other hilarine genera (personal observation), but no significant sclerotization of the posterior margin is clearly observed. However, this remains to be confirmed by studying a representative sample of hilarine species belonging to various genera. The third characteristic to take into consideration is the presence of setae on the ventral surface of vein R_2+3_. This last characteristic is also found in the monotypic Chilean genus, *Pasitrichotus*.

*Tropicohilara* **gen. nov.** and *Pasitrichotus* share the following characteristics: a thin basal fore tarsomere (sometimes present in species belonging to other genera) and the absence of a set of antero-apical setae on the fore tibia. However, the only characteristic seemingly exclusive to this group is the presence of setae on the ventral vein R_2+3_. In contrast, *Pasitrichotus* exhibits some unique characteristics that set it apart. For instance, *Pasitrichotus* exhibits a distinct ventral row of spines on the mid tibia and possesses setae on both R_1_ and R_4+5_ veins, both characteristics are absent in *Tropicohilara* **gen. nov.**

*Tropicohilara* **gen. nov.** has hairs on the face, along the margin of the eyes (absent in *Pasitrichotus*). The presence of hairs on the face is not exclusive to this genus. Many species of *Hilarigona* Collin, 1933, some species of *Hilarempis*, and *Hilara* also exhibit these facial hairs.

In *Tropicohilara* **gen. nov.**, the hind tibia is shorter than the hind femur, a characteristic that is also found in species of *Aplomera* Macquart, 1839 and *Hilarigona*, while *Pasitrichotus* has hind tibia length equal to hind femur.

In summary, *Tropicohilara* **gen. nov.** and *Pasitrichotus* share certain morphological characteristics, such as the thin basal fore tarsomere and the absence of antero-apical setae on the fore tibia. However, they also exhibit distinct traits unique to each genus, such as the presence of setae on specific wing veins for *Pasitrichotus* and the triangular-shaped occiput and shorter hind tibia for *Tropicohilara* **gen. nov**. These differences and similarities can be useful for taxonomic classification and evolutionary studies within these genera.

## Data Availability

Data is contained within the article.

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
