# Peer review of "Tropicohilara, a New Genus of Hilarini (Diptera: Empididae: Empidinae) from Brazil, with Descriptions of Six New Species†"

_insects, 2023, doi:10.3390/insects14120915_

Round 1

Reviewer 1 Report

Comments and Suggestions for Authors

This is a very nice work - comprehensive and thorough with excellent illustrations. I have made some comments on the manuscript in Track Change (Word version).

My copy of the manuscript was missing figures 6 and 7.

Author Response

All comments have been taken into account and the paper has been revised accordingly. in particular some references were added and references updated

Comment A7: we deleted the comparison with Aplomera, which is not relevant.

Reviewer 2 Report

Comments and Suggestions for Authors

The paper provides interesting material on the family Empididae (dance flies) from  South America, where this group remains poorly studied. The authors describe a new genus of the tribe Hilarini comprising six new species from the Amazon and Atlantic Forest biomes (Brazil). In addition, a key to these species is provided. The contribution of the paper is new and original. The keywords and abstract clearly indicate the content of the paper. The descriptions of the new species are adequate, concise and clear; they are well differentiated and the International Code of Zoological Nomenclature is respected. Also, the descriptions of the new species addresses all relevant issues (etymology, deposition of type material, localisation of type and other localities, measurements, differential diagnosis, description). The descriptions are backed up with very good photos of the habitus and the crucial morphological features. The list of references is relatively comprehensive (but see remarks in the text).

Remark:

The authors should somewhat upgrade the list of references to include some recent publications (see text).

Author Response

All comments have been taken into account and the paper has been revised accordingly. 

References have been updated. In particular we added several recent publications.

Reviewer 3 Report

Comments and Suggestions for Authors

This manuscript describes a new genus from the highly diverse and morphologically diverse genus complex of Hilarempis. The authors present a good argument justifying their decision and it definitely appears to represent a monophyletic lineage. Most importantly it appears that females are also identifiable. I have made extensive comments/edits on the Word version of the manuscript. Many of these are created by the journal format. Special care is required by the authors to improve the presentation product. 

I also have the following points:

1. The plates for Figs 6 and 7 were not included in this review.

2. A slide mount or flattened wing figure would be an improvement in the description of the wing.

3. In the genus description, please add that R1 is bare. This is the most often vein to be setose in empidids and it is rather unusual that this vein is bare and R2+3 is setose.

4. A key to South American hilarine genera would be useful, or at least where this new genus would key in Collin (1933).

Comments on the Quality of English Language

Only minor edits to the English are required and this has mostly been noted in the attached annotated Word version of the manuscript.

Author Response

Comment Rev1: The registration of the paper is in progress and should be ready for publication.

Comment Rev5 : We did not include a key to hilarine genera. The combination of characters given in the diagnosis (which was improved) makes it easy to recognise the new genus. In addition, we indicate in the discussion the genera with which Tropicohilara should be compared.

Comment Rev12 : the costs folded over

Comment Rev15 : yes we agree, and at this stage we can say no more.

We did not include a slide of wing but we added an explanation of veins and cells in the figure 8.

The remaining comments have been taken into account and the paper has been revised accordingly.

Reviewer 4 Report

Comments and Suggestions for Authors

General remark: There is no consistency in using italics for genera and species, and for the abbreviations „gen. nov.“ and „sp. nov.). The latter are sometimes written in italics as well (e. g. lines 89, 255), or italics are avoided for genus names (e. g. 642) [I did not check the file line for line, this should be done by the authors to reach consistency].

In point 2 Materials and methods a line should be added how the sections “Type material” should be written. What is the meaning of the quotation marks etc.

As to the figures: I would recommend adding an explanation of veins and cells at the wing at least in one figure (e.g. Fig. 8E).

Lines 44 to 49: In my point of view, here should be added, that in the (probably) highest derived species the nuptial gifts do not contain „insects“ (prey) any longer but consist only of silk (e. g. Hilara sartor, Europe).

Lines 72 to 74: were the collecting „items“ only single units? Malaise trap => Malaise traps (?) etc.

Line 92 to 94: [Etymology]. The name „Tropicohilara“ does not contain a hint to the South American tropical region, hence the explanation should be re-written. The explanation of „Tropics“ should explain that this region is situated between the „Tropic of Cancer“ and the „Tropic of Capricorn“ around the whole globe with the aequator in its center.

Lines 155 to 172: I would recommend adding the years of description of the genera mentioned in this paragraph. On the other hand the remarks in lines 169 to 172 belong to the „Discussion“ (beginning with line 570). All in all, I would recommend a more precise differential diagnosis as far the new genus is considered. (Are there other Hilarini with predominat yellow colour of the body including the legs etc.).

[e.g.] Line 208: I have not checked every line in each species, but here it is evident, that there is a mixture of SI-units used in „words“ => „mts“ is probably „meters“; the abbreviation should be „m“ only. (Normally the abbreviation „m a.s.l.“ should be used => meters above sea level. (I see, that this is the citation of original label).=> see my remark to point 2 Material and methods

Lines 584 to 586: There should at least be one hint to the shape of the heads as shown in the figures (e. g.: „Best seen in fig xxx“). What is meant with „more or less horizontal“? (line 585).

[Here I allow me to add some further remarks in general. I know, that within a taxonomic paper with the main aim to describe a new genus with several new species it is sometimes not possible to give a complete statement on the phylogeny of the genus in question. Nevertheless, I have the imagination (and may be other workers on Empidoidea would have the same feeling, while checking the new genus), that head and thorax (aside from the Hilarini character, [a bare laterotergite] and the morphology of the hypopygium etc.) are very similar to Empis (Xanthempis). If only the head is considered its shape is comparable to that of Xanthempis. There is the phenomenon of heterobathmic development of characters in organisms. That means, that single characters in a given mosaic of characters have a faster evolutionary development than others and – vice versa – in other groups these characters are in stasis. It is the question, if the obviousely similarity of heads (with the antenna! => prolonged scapus!) is a parallel development or has its base in a common early state. I would find it useful, to give a statement on this matter in the discussion. On the other hand I strongly would recommend to cite at least one paper of Dan Bickel, in which he states his opinion on megadiverse taxa within the Empidoidea. The authors give their opinion, that there are „many“ species undescribed at the moment. In the Australasian Realm Bickel states a real megadiversity of Hilarini (which leads him to the opinion, that it would be nearly impossible, to describe all species with the necessary thorougness …).   

Comments on the Quality of English Language

Minor faults detected. Please check my comments.

Author Response

All comments have been taken into account and the paper has been revised accordingly.

The discussion of the morphological similarity between Tropicohilara and Xanthempis and the evolution of certain characters has not been included in the manuscript. Although interesting, as the reviewer pointed out, the paper is not at all focused on this theme and we considered that adding a discussion on this topic was beyond the scope of the paper and of the revision  for publication.